# Extraction of Added-Value Triterpenoids from *Acacia dealbata* Leaves Using Supercritical Fluid Extraction

**Vítor H. Rodrigues** [ID]**, Marcelo M. R. de Melo, Inês Portugal** [ID] **and Carlos M. Silva \*** [ID]

CICECO—Aveiro Institute of Materials, Department of Chemistry, University of Aveiro, Campus Universitário de Santiago, 3810-193 Aveiro, Portugal; vitorhrodrigues@ua.pt (V.H.R.); marcelo.melo@ua.pt (M.M.R.d.M.); inesport@ua.pt (I.P.)
**\*** Correspondence: carlos.manuel@ua.pt

**Abstract:** Forestry biomass is a by-product which commonly ends up being burnt for energy generation, despite comprising valuable bioactive compounds with valorisation potential. Leaves of *Acacia dealbata* were extracted for the first time by supercritical fluid extraction (SFE) using different conditions of pressure, temperature and cosolvents. Total extraction yield, individual triterpenoids extraction yields and concentrations were assessed and contrasted with Soxhlet extractions using solvents of distinct polarity. The extracts were characterized by gas chromatography coupled to mass spectrometry (GC-MS) and target triterpenoids were quantified. The total extraction yields ranged from 1.76 to 11.58 wt.% and the major compounds identified were fatty acids, polyols, and, from the triterpenoids family, lupenone, α-amyrin and β-amyrin. SFE was selective to lupenone, with higher individual yields (2139–3512 mg $kg^{-1}_{leaves}$) and concentrations (10.1–12.4 wt.%) in comparison to Soxhlet extractions, which in turn obtained higher yields and concentrations of the remaining triterpenoids.

**Keywords:** *Acacia dealbata*; GC-MS; leaves; lupenone; supercritical fluid extraction; Soxhlet extraction; triterpenoids

## 1. Introduction

The genus *Acacia* is widespread through the Portuguese landscape, consisting of three main species: *Acacia dealbata*, *Acacia longifolia* and *Acacia melanoxylon* [1]. *A. dealbata* was introduced for dune erosion protection as well as ornamental and wood supply purposes during the 19th and 20th century [2]. Currently, it is considered a plague due to its fast growth and dominance over the natural flora [1,3]. From 2005 to 2015, the occupied area of *Acacia* species increased 4000 ha in Portugal, corresponding to an estimated total arboreal biomass growth of 2 Mt [4]. The removal of these trees generates forest biomass that, under the Renewable Energy Directive II of the European Union Commission [5], can be utilized for the production of liquid and gaseous biofuels. However, it is a common practice to leave these residues in the forest for soil remediation.

The research towards *A. dealbata* biomass extraction has focused on several morphological parts, namely wood [6–10], bark [6–8,10–14], flowers [11,15–20] and leaves [6,8,11,17,21,22]. The explored extraction methods so far consist of solid-liquid extraction with organic solvents, such as dichloromethane, ethanol, methanol, hexane, acetone and some hydroalcoholic mixtures. Extraction of essential oils by steam distillation has been applied only to flowers [16]. Besides these conventional methods, there are few works on greener and more innovative extraction procedures, such as the work of Borges et al. [22], who applied microwave and ultrasound-assisted extraction to the leaves, and Lopez-Hortas et al. [16], using microwave hydrodiffusion to obtain the flower essential oil. One alternative technique for the extraction of vegetable biomass is supercritical fluid extraction (SFE) [23]. It is mainly employed with carbon dioxide ($CO_2$) as solvent due to its low cost, safety,

availability and low critical point conditions, which allows extraction at near room temperatures [24]. The manipulation of temperature and pressure allows the tuning of $CO_2$ properties (density, viscosity and diffusivity) that maximize the desired responses, such as total extraction yield or the selective uptake of target chemical families or compounds. For example, triterpenic acids in the case of *Eucalyptus globulus* leaves [25,26] and bark [27–29], triterpenes from *Vitis vinifera* leaves [30], friedelin from *Quercus cerris* cork [31] and sterols from *Eichhornia crassipes* [32].

The phytochemistry of *A. dealbata* biomass (bark, leaves, wood, flowers and seeds) includes several families of compounds, such as alkaloids [8], amines [33], phenolics [8,10,13,15], polysaccharides [9], chalcone glycosides [18], steryl glucosides [7], tannins [10,11,13], caffeic acid esters [12], sterols [6] and triterpenes [6,17]. SFE is a proven technology for the selective removal of triterpenes and sterols from many vegetable matrices [23,30–32,34,35]. For instance, compounds such as lupenone, lupeol, lupenyl palmitate, lupenyl cinnamate, squalene, β-amyrone, α-amyrin, β-amyrin and 22,23-dihydrospinasterol have already been identified and quantified [6,17]. These have been reported for several potential bioactive properties, namely anti-inflammatory, anti-virus, anti-diabetes, anti-cancer and antiproliferative, among others [36–45] which may explain the association of *Acacia* species with traditional medicine practices [46,47]. The wide range of biological activities potentiates the interest for multiple applications of the extracts, for example, to obtain active pharmaceutical ingredients or for incorporation in nutraceuticals, food, animal feed and cosmetic products.

This work focuses on the SFE of triterpenoids from *Acacia dealbata* leaves under different experimental conditions of pressure, temperature and cosolvents content, and its comparison with conventional Soxhlet extraction using organic solvents of distinct polarity. The extracts were characterized by gas chromatography coupled to mass spectrometry (GC-MS) and triterpenoids contents were determined. To the best of our knowledge, this is the first time SFE is applied to *Acacia dealbata* leaves aiming for the extraction of potential bioactive compounds.

## 2. Materials and Methods

### 2.1. Chemicals

Carbon dioxide ($CO_2$, purity 99%) was supplied by Air Liquide (Algés, Portugal). Dichloromethane (purity 99.98%), *n*-hexane (purity 99%) and ethanol (purity 99.5%) were supplied by Fisher Scientific (Leicestershire, UK). Ethyl acetate (purity 99%) was supplied by VWR International (Fontenay-sous-Bois, France). Pyridine (purity 99.5%), tetracosane (purity 99%) N,O-Bis(trimethylsilyl)trifluoroacetamide (BSTFA, purity 98%) and chlorotrimethylsilane (TMSCl, purity 99%) were supplied by Sigma Aldrich (Madrid, Spain). Betulinic, oleanolic and ursolic acids (purity 98%) were supplied by AK Scientific (Union City, CA, USA).

### 2.2. Acacia Dealbata Biomass

The *A. dealbata* leaves were supplied by RAIZ—Forest and Paper Research Institute (Eixo, Portugal). The leaves were collected from 8-year-old *Acacia dealbata* trees located in Porto/Valongo (Portugal) region during the winter season. The leaves (see Figure 1) were manually separated from the branches and dried at 35 °C for 72 h in a forced convection oven, reducing their moisture content from 65.6 to 4.5 wt.%.

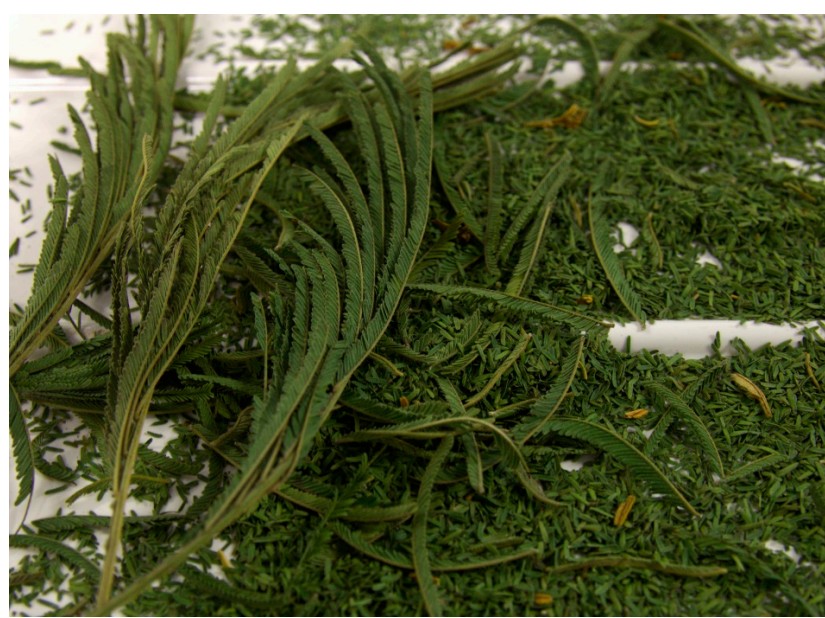

**Figure 1.** Oven-dried *Acacia dealbata* leaves.

### 2.3. Soxhlet Extraction

The leaves of *A. dealbata* were extracted with *n*-hexane, dichloromethane, ethyl acetate and ethanol (Table 1). In each assay, extraction of the leaves (ca. 3 g) was performed with 180 mL of each solvent for 6 h. The produced extracts were evaporated to dryness in a rotary evaporator, weighed for the determination of total extraction yield ($\eta_{\text{Total}}$, wt.%) and analyzed by GC-MS to evaluate triterpenoids individual yields ($\eta_i$, mg $kg_{\text{leaves}}^{-1}$) and concentrations ($C_i$, wt.%), as follows:

$$\eta_{\text{Total}} = \frac{m_{\text{extract}}}{m_{\text{dry leaves}}} \times 100 \tag{1}$$

$$\eta_i = \frac{m_i}{m_{\text{dry leaves}}} \times 10^6 \tag{2}$$

$$C_i = \frac{m_i}{m_{\text{extract}}} \times 100 \tag{3}$$

where $m_{\text{dry leaves}}$ is the mass of dry leaves, $m_{\text{extract}}$ corresponds to extract mass free of solvent and $m_i$ is the mass of triterpenoids measured by GC-MS.

**Table 1.** List of Soxhlet and SFE assays with the respective operating conditions.

| Run | Method | Solvent | T (°C) | P (bar) | $\rho_f$ (kg m$^{-3}$) |
|---|---|---|---|---|---|
| SX1 | Soxhlet | *n*-Hexane | 68.5 * | 1 | - |
| SX2 | | Dichloromethane | 39.6 * | 1 | - |
| SX3 | | Ethyl acetate | 77.1 * | 1 | - |
| SX4 | | Ethanol | 78.4 * | 1 | - |
| SFE1 | SFE | $CO_2$ | 40 | 200 | 840.6 [48] |
| SFE2 | | $CO_2$ | 80 | 200 | 594.9 [48] |
| SFE3 | | $CO_2$ | 60 | 300 | 830.4 [48] |
| SFE4 | | $CO_2$:Ethanol (95:5 wt.%) | 80 | 300 | 764.6 [49] |
| SFE5 | | $CO_2$:Ethyl acetate (95:5 wt.%) | 80 | 300 | 761.4 [50] |

* boiling point of the pure solvents.

### 2.4. Supercritical Fluid Extraction

The SFE assays were performed in a lab scale Spe-ed SFE unit, a model of Helix SFE System-Applied Separations, Inc., (Allentown, PA, USA) schematically presented in Figure 2. In each run, ca. 25 g of leaves were loaded into the extractor while the supercritical fluid flowed upwards at constant flow rate ($Q_{CO_2}$) of 12 g min$^{-1}$ for 6 h. The experimental conditions of pressure, temperature and cosolvent content are presented in Table 1 (runs SFE1 to SFE3). The detailed procedure is described elsewhere [25]. The total extraction yield, the individual compound concentrations and respective yields were determined according to Equations (1)–(3), respectively.

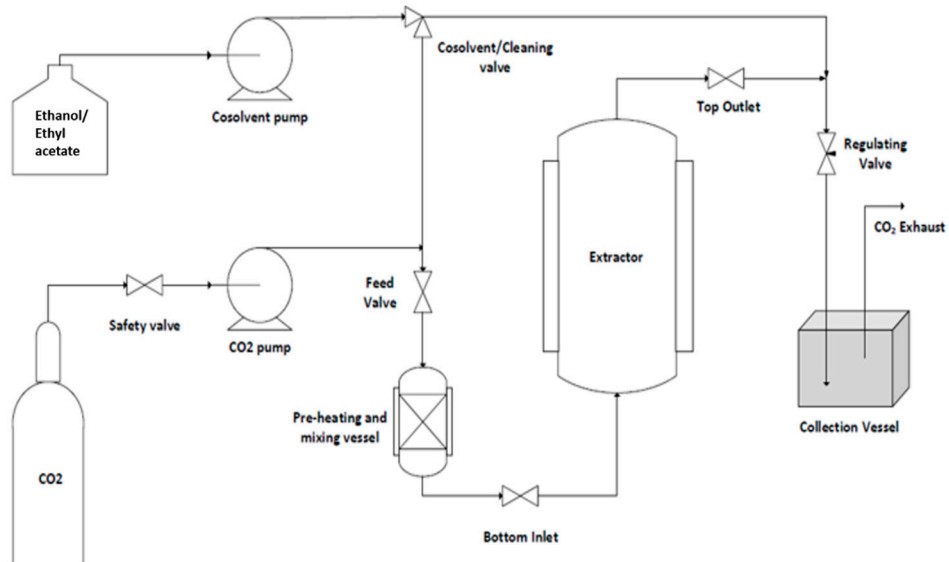

**Figure 2.** Simplified scheme of the SFE installation. Reprinted with permission from [25]. Copyright 2021, Elsevier.

For the runs, SFE4 and SFE5 ethanol and ethyl acetate were added as cosolvent to modify the supercritical fluid polarity and the solubility of solutes. In these runs, the cosolvent was fed to the pre-heating vessel using a HPLC pump, as presented in Figure 2.

The densities of the supercritical fluids ($\rho_f$), both pure and modified supercritical carbon dioxide (SC-CO$_2$), at each experimental condition are presented in Table 1. They were obtained using the equation of state of Pitzer and Schreiber for pure SC-CO$_2$ [48], Falco and Kiran for SC-CO$_2$ modified with ethyl acetate [50] and Pöhler and Kiran for SC-CO$_2$ modified with ethanol [49].

### 2.5. Gas Chromatography Coupled to Mass Spectrometry

The extracts were analyzed by GC–MS using a Trace Gas Chromatograph Ultra equipped with a DB-1 J&W capillary column (30 m × 0.32 mm i.d., 0.25 μm film thickness) and coupled with a Thermo DSQ mass spectrometer. The extracts were prepared and analyzed following a procedure previously published [28,29]. For the quantification of individual triterpenoids in the extracts, tetracosane and pure betulinic, oleanolic and ursolic acids were selected as internal and external standards, respectively. The identification of the compounds was performed with the aid of reverse match factors (RSI) from Wiley 9 library, which defines thresholds of mass spectral match: 900 and above is considered excellent; 800–900 is considered good, 700–800 is considered fair, and below 700 is considered a poor match [51].

## 3. Results and Discussion

### 3.1. Total Extraction Yield

The total extraction yield ($\eta_{Total}$, Equation (1)) measures the total extract amount produced independently of its composition. The results obtained with Soxhlet and SFE can be visualized in Figure 3. Concerning Soxhlet extraction, the $\eta_{Total}$ values increased with the polarity of the solvent and ranged from 3.60 to 11.58 wt.% for dichloromethane and ethanol, respectively. The second highest value (7.97 wt.%) was obtained by ethyl acetate, and *n*-hexane achieved an equivalent value to dichloromethane (3.64 wt.%). Literature results on Soxhlet extractions of *A. dealbata* leaves present unequal yield scores. For instance, Oliveira et al. [6] obtained a $\eta_{Total}$ of 6.2 wt.% with dichloromethane, almost double of the value obtained in this work, which may be due to the particle size reduction performed. On the other hand, Luís et al. [8] obtained a $\eta_{Total}$ of 6.75 wt.% with ethanol, which is considerably lower than the value obtained in this work, and may be due to a different time of extraction, as it was stopped as soon as the extraction solvent became colorless. Borges et al. [22] obtained a $\eta_{Total}$ of 13 wt.% using water and 16 h of Soxhlet extraction, which can compare with the value obtained with ethanol (11.58 wt.%), the closest solvent in terms of polarity. However, it is noteworthy that extraction with ethanol (6 h at 78.4 °C) would be more energy efficient than with water (16 h at 100 °C).

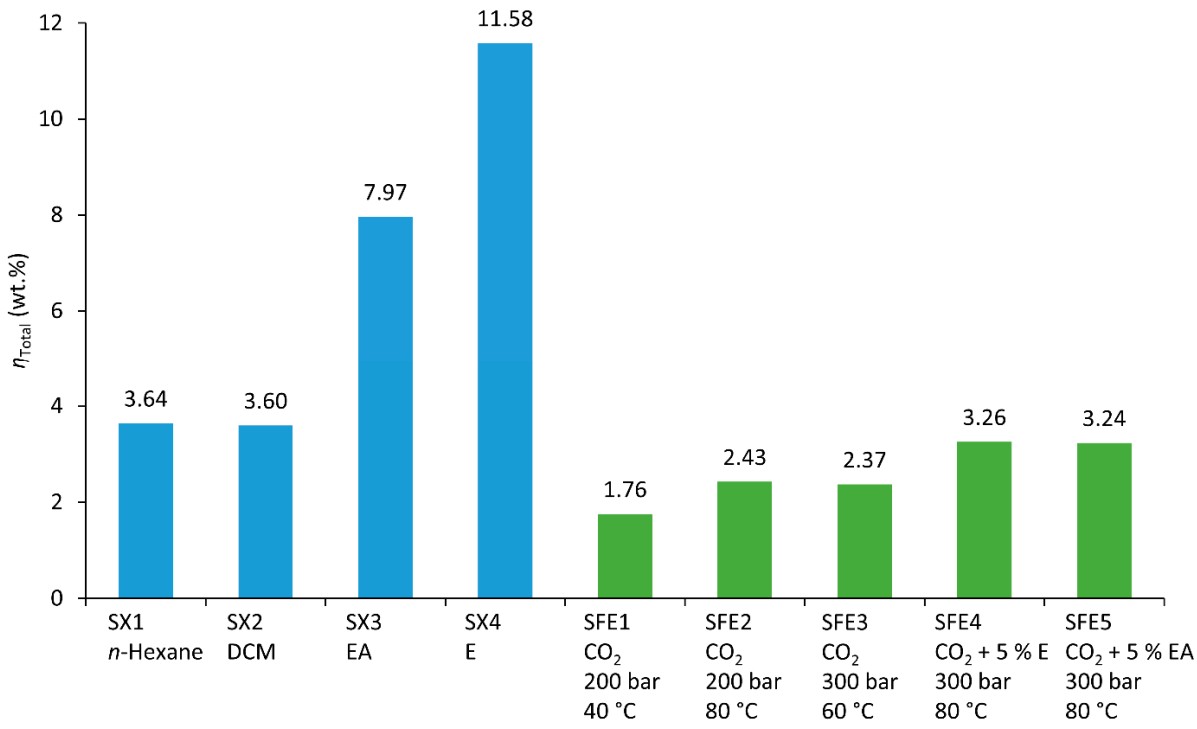

**Figure 3.** Total extraction yields ($\eta_{Total}$) obtained by Soxhlet using *n*-hexane, dichloromethane (DCM), ethyl acetate (EA), ethanol (E) and SFE at different conditions.

Regarding SFE assays, the $\eta_{Total}$ results varied from 1.76 to 3.26 wt.% for SFE1 (200 bar, 40 °C, no cosolvent) and SFE5 (300 bar, 80 °C, 5 wt.% ethanol), respectively. Three experimental parameters were tested in these runs: pressure, temperature and cosolvent addition. Setting SFE1 (200 bar, 40 °C, no cosolvent) as the reference run, the effect of temperature on $\eta_{Total}$ can be assessed by comparison with run SFE2 (200 bar, 80 °C, no cosolvent), where an increase of 38% was observed. Even though higher temperatures lower the solvent power due to SC-$CO_2$ density decrease (840.6 kg m$^{-3}$ for SFE1 and 594.6 kg m$^{-3}$ for SFE2), they increase the vapour pressure of solutes (i.e., their solubility in the supercritical solvent). These opposing effects can lead to different results from system to system, and in this case, the solubility enhancement effect was prevalent. Moving to Run SFE3 (300 bar, 60 °C, no

cosolvent), this assay was performed at more 100 bar and less 20 °C of SFE2, but yielded an identical $\eta_{Total}$ (2.37 wt.%). This occurs possibly because the higher SC-CO$_2$ density in Run 3 (830.4 kg m$^{-3}$, i.e., 40% more than in SFE2) compensated the thermal penalization on the vapor pressure side of solutes. Furthermore, the addition of cosolvents was tested at 300 bar and 80 °C in runs SFE4 (5 wt.% of ethanol) and SFE5 (5 wt.% of ethyl acetate). The results were identical for both runs (3.24–3.26 wt.%), almost doubling the $\eta_{Total}$ value of SFE1. The different cosolvents did not significantly affect the fluid densities between each other (764.6 kg m$^{-3}$ for SFE4 and 761.4 kg m$^{-3}$ for SFE5) and imposed an increase of only 2.4% in relation to pure SC-CO$_2$ under the same $P - T$ conditions (746.2 kg m$^{-3}$) [48]. Hence, the yield gain in runs SFE4 and SFE5 was attributed to a greater affinity of the solute to solvent mixtures of higher polarity.

Overall, it is clear that Soxhlet extraction produces higher $\eta_{Total}$, especially when employing high polarity solvents (i.e., ethanol or ethyl acetate). When employing low polarity solvents (i.e., dichloromethane or *n*-hexane), the $\eta_{Total}$ values are analogous to those of modified SC-CO$_2$ and higher than those of SFE without entrainers.

*3.2. Volatile Extractives*

The extracts were analyzed by GC-MS, and the chromatograms of runs SX2 (A), SX4 (B) and SFE1 (C) are presented in Figure 4A–C. These are representative of the Soxhlet extractions with low and higher polarity solvents, and SFE runs, respectively. It is possible to observe that the chromatograms of runs SX2 and SFE1 (Figure 4A,C) are considerably similar, as the same peaks appear in both, which in turn confirms the affinity of dichloromethane and SC-CO$_2$ to similar solutes. On the contrary, SX4 (Figure 4B) stands out due to the proliferation of peaks in the region on the left of the internal standard—i.e., at retention time (Rt) lower than 38.76 min—and also due to the very sharp peak at Rt = 27.55 min, identified as *myo*-inositol (whose structure is disclosed in Figure 5). The latter forced a rescaling of the relative absorbance axis to a comparable range (from 100% in SX2 and SFE1 to 6% in SX4). All but one of the identified compounds in these peaks encompassed RSI scores comprehended between 700 and 942, which correspond to a matching quality from fair to excellent. In fact, the only exception to this was α-amyrin, whose RSI score was 691.

The full list of identified compounds for all Soxhlet extractions and SFE runs is reported in Table 2, altogether with the respective retention times and maximum RSI scores within the analyzed extracts. As previously observed in Figure 4B, run SX4 shows more peaks in the left half of the chromatogram, and that observation can be confirmed in Table 2, also for runs SX3 (ethyl acetate) and SX4 (ethanol), specifically amid retention times of 10.77 to 44.37 min. The said peaks consist mainly of polyols (P) and monosaccharides (M). After these come the fatty acids (FA), long-chain aliphatic alcohols (LCAA) and triterpenoids (TT), although these were also found in every extract. Furthermore, SX1 (*n*-hexane) and SX2 (dichloromethane) present a very similar pattern of detected compounds, as well as the runs SFE1-SFE5. This corroborates what was observed in the analysis of Figure 4A,C. Such similarities show that the polarity of the solvent strongly influences the compounds extracted, and that, in the case of SFE, the modification of SC-CO$_2$ with 5 wt.% of ethanol or ethyl acetate was not enough to lead to the extraction of volatile compounds with higher affinity to polar organic solvents.

**Figure 4.** Chromatograms of the extracts from runs (**A**) SX2 (dichloromethane), (**B**) SX4 (ethanol) and (**C**) SFE1 (200, bar 40 °C). Internal standard (tetracosane) appears at 38.7 min.

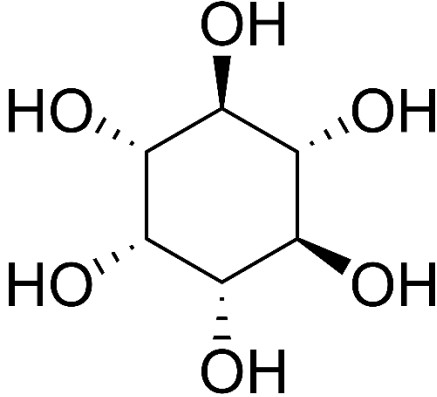

**Figure 5.** Structural formula of *myo*-inositol.

**Table 2.** List of identified compounds and respective retention times and reverse match factors (RSI) for all Soxhlet and SFE runs.

| Rt (min) | Compound | Family | RSI | SX1 | SX2 | SX3 | SX4 | SFE1 | SFE2 | SFE3 | SFE4 | SFE5 |
|---|---|---|---|---|---|---|---|---|---|---|---|---|
| 10.77 | Glycerol | P | 909 | - | + | + | + | - | - | - | - | - |
| 18.67 | Erythritol | P | 942 | - | - | + | + | - | - | - | - | - |
| 20.6 | 5-Hydroxypipecolic acid | CA | 770 | - | - | - | + | - | - | - | - | - |
| 24.17 | Xylitol | P | 800 | - | - | + | + | - | - | - | - | - |
| 24.79 | Ribitol | P | 887 | - | - | + | + | - | - | - | - | - |
| 27.55 | *myo*-Inositol | P | 803 | - | - | + | + | - | - | - | - | - |
| 28.19 | Tyramine | A | 890 | - | - | - | + | - | - | - | - | - |
| 28.79 | D-Mannose | M | 803 | - | - | + | + | - | - | - | - | - |
| 30.13 | Mannitol | P | 851 | - | - | + | + | - | - | - | - | - |
| 31.27 | Glucose | M | 827 | - | - | + | + | - | - | - | - | - |
| 44.37 | Docosanoic acid | FA | 729 | + | + | - | - | + | + | + | + | + |
| 46.09 | Squalene | TT | 833 | - | - | - | - | + | + | + | + | + |
| 47.76 | Pentacosanoic acid | FA | 829 | + | + | + | + | + | + | + | + | + |
| 48.61 | Hexadecanoic acid | FA | 705 | + | + | + | - | - | - | - | - | - |
| 50.98 | Octacosanoic acid | FA | 849 | + | + | + | + | + | + | + | + | + |
| 51.82 | Octacosan-1-ol | LCAA | 811 | + | - | + | - | - | - | - | - | - |
| 52.66 | 4′-OH,5-OH,7-Di-O-Glucoside | F | 688 | + | + | + | + | + | + | + | + | + |
| 52.86 | α-Amyrone | TT | 786 | + | + | - | - | + | + | + | + | + |
| 53.06 | β-Amyrone | TT | 738 | + | + | + | + | + | + | + | + | + |
| 53.66 | Lupenone | TT | 843 | + | + | + | + | + | + | + | + | + |
| 54.92 | β-Amyrin | TT | 755 | + | + | + | + | + | + | + | + | + |
| 55.16 | α-Amyrin | TT | 691 | + | + | + | + | + | + | + | + | + |
| 56.28 | Lupenyl acetate | TT | 742 | + | + | - | - | + | + | + | + | + |

P—polyol; CA—carboxylic acid; A—amine; M—monosaccharide; FA—fatty acid; TT—triterpenoid; LCAA—long-chain aliphatic alcohol; F—flavonoid.

As observed in Figure 4B, *myo*-inositol (see Figure 5) peak has an area of different magnitude from the targeted triterpenoids, thirteen times higher (run SX4) than the internal standard. Even though it is a known constituent of plant and animal cells, the results obtained demonstrate the potential of ethanol and ethyl acetate, and potentially other polar organic solvents, for the production of *myo*-inositol-rich extracts. According to the literature, this compound and its derivatives have been identified and quantified in several plant species [52–56], including *Acacia* trees, such as in *Acacia pennata* and *Acacia farnesiana* leaves [57], and *Acacia mangium* and *Acacia maidenii* seeds [58]. Moreover, *myo*-inositol plays an important role in several cell functions, such as growth, development and reproduction, among others [59]. As a dietary supplement, it can be beneficial for human disorders associated with insulin resistance, such as polycystic ovary syndrome, gestational diabetes mellitus or metabolic syndrome, and the prevention or treatment of some diabetic complications, namely, neuropathy, nephropathy and cataract [59]. Even though this represents a promising result for the valorisation of *Acacia dealbata* biomass, the main focus of this work is the triterpenoid fraction attainable by SFE, which leaves *myo*-inositol out of the work scope. Nevertheless, it might open the way to sequential extraction strategies.

The main triterpenoids (TT) identified in the produced extracts were squalene, α-amyrone, β-amyrone, lupenone, β-amyrin, α-amyrin and lupenyl acetate. These contain 30 carbon atoms, except for lupenyl acetate which has 32, and all of these compounds are interrelated by known biosynthesis pathways, as summarized in Figure 6. Here, it can be seen that squalene is the prime precursor, having been originated by successive condensation reactions of the isomers of isopentenyl diphosphate and dimethylallyl diphosphate [60–62]. Eventually, squalene can be oxidized to 2,3-oxidosqualene, which in turn is the direct precursor of tricyclic, tetracyclic or pentacyclic triterpenoids. When under the chair-chair-chair conformation, 2,3-oxidosqualene can undergo cyclization reactions forming the tetracyclic dammarenyl cation, which, after ring expansions, originates diverse

skeletons of pentacyclic triterpenoids, such as the lupane, oleanane and ursane types (see Figure 6). The latter three types are the precursors of lupeol, β-amyrin and α-amyrin, respectively [60–63]. Upon undergoing further rearrangements, and/or oxidation, substitution or glycosylation reactions, other triterpenoids are generated, namely lupenone, β-amyrone and α-amyrone (as the ketone versions of lupeol, β-amyrin and α-amyrin, respectively), and lupenyl acetate (as the acetylated version of lupeol). To conclude, the majority of these compounds were also identified in previous extraction works of *A. dealbata* biomass [6,17], especially in the leaves, bark and other external parts, since they are thought to provide protection against insects and microbes [63].

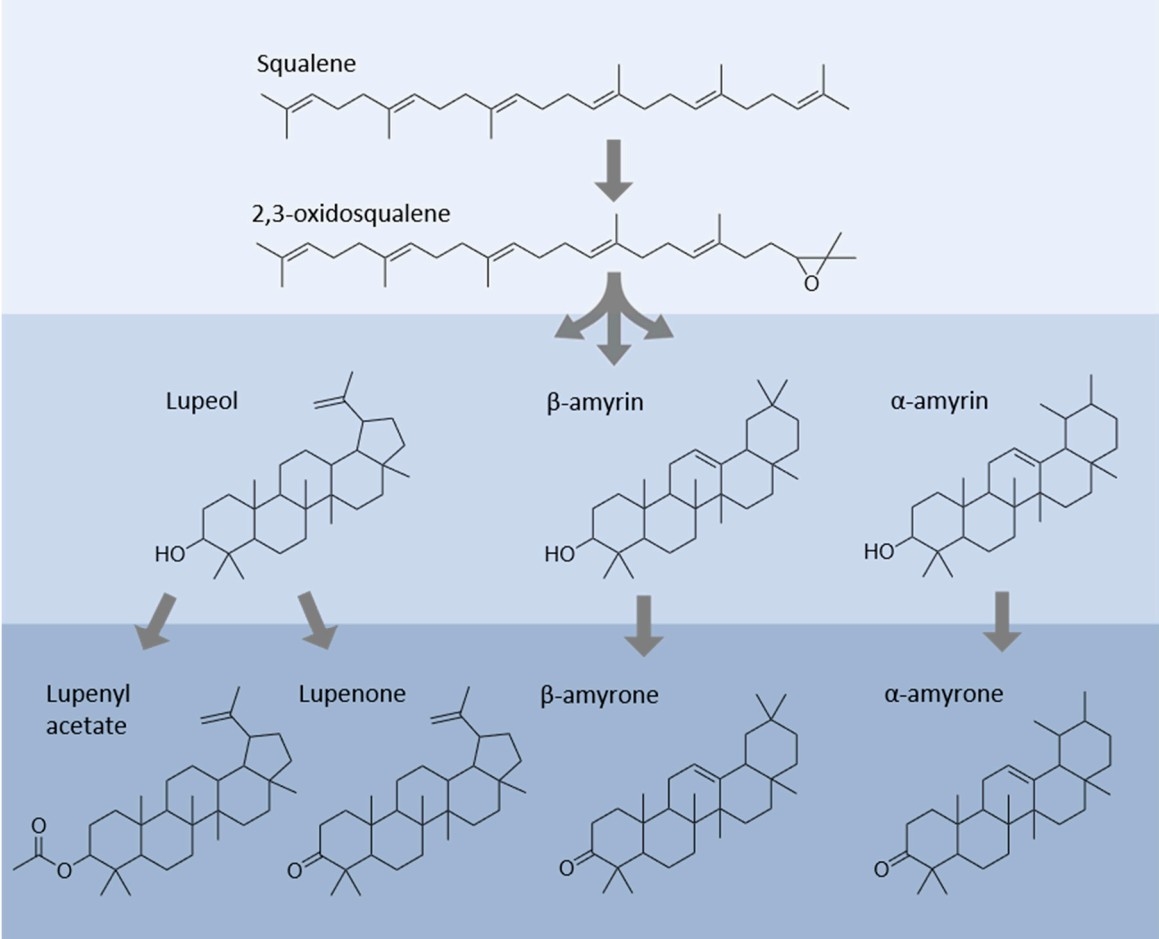

**Figure 6.** Scheme representative of synthesis pathways of triterpenoids from squalene, namely the lupane, oleanane and ursane series.

### 3.3. Triterpenoid Extraction Yields

The individual ($\eta_i$) and total triterpenoid ($\eta_{\text{Total TT}}$) extraction yields of Soxhlet and SFE assays are presented in Figure 7A,B, respectively. Regarding Soxhlet extractions (Figure 7A), lupenone was the most extracted compound of the four organic solvents tested, with $\eta_{\text{lupenone}}$ values ranging from 2114 to 2994 mg $\text{kg}^{-1}_{\text{leaves}}$ for *n*-hexane and ethyl acetate extraction, respectively. It was followed by α-amyrin, with yields from 1249 to 2851 mg $\text{kg}^{-1}_{\text{leaves}}$ for dichloromethane and ethyl acetate, respectively. Although $\eta_{\beta-\text{amyrin}}$ was generally lower than $\eta_{\alpha-\text{amyrin}}$, this difference is especially evident in the dichloromethane Soxhlet extract, where the latter yielded circa 2.7 times more. In turn, ethyl acetate attenuated the two amyrin yields, with their ratio falling to ca. 1.6. For extracts produced with more polar solvents, the two amyrin yields approached the values of $\eta_{\text{lupenone}}$. This was translated to the magnitude of $\eta_{\text{Total TT}}$, which incremented from the minimum of 4908 mg $\text{kg}^{-1}_{\text{leaves}}$ for *n*-hexane to 8201 mg $\text{kg}^{-1}_{\text{leaves}}$ for ethyl acetate and

to 6259 mg $kg^{-1}_{leaves}$ for ethanol. The remaining triterpenoids did not differ significantly between organic solvents, and their individual yields were markedly low.

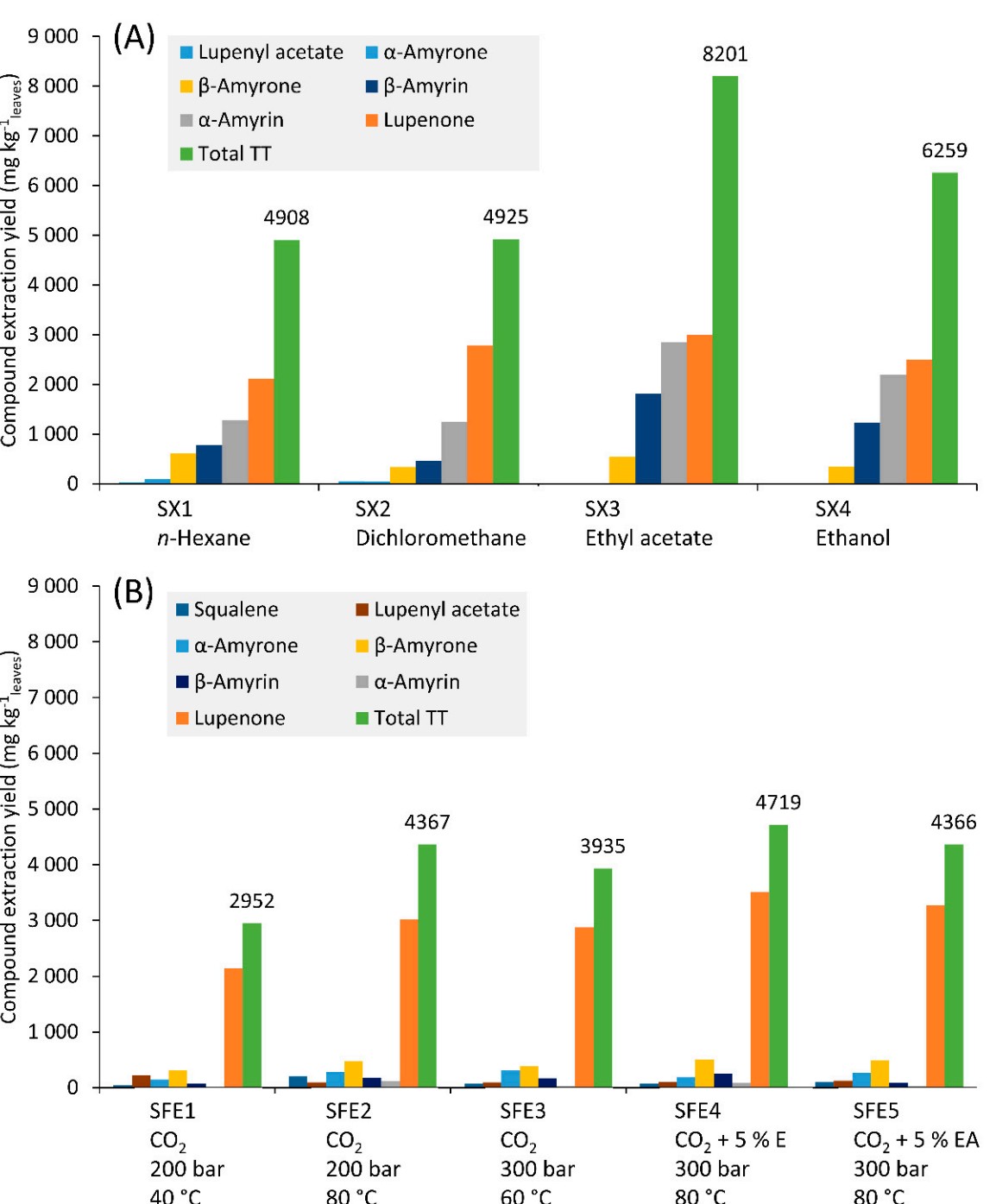

**Figure 7.** Individual and total triterpenoid (Total TT) extraction yields in: (**A**) Soxhlet extracts using different organic solvents, and (**B**) SFE at different conditions of temperature, pressure, and ethanol (E) or ethyl acetate (EA) content as modifiers.

Concerning uptake of triterpenoids by SFE (see Figure 7B), lupenone confirmed its leading individual yield also in this separation method, which varied from 2139 to 3512 mg $kg^{-1}_{leaves}$ for SFE1 (200 bar, 40 °C, no cosolvent) and SFE4 (300 bar, 80 °C, 5 wt.% of ethanol), respectively. None of the remaining triterpenoids surpassed 500 mg $kg^{-1}_{leaves}$ threshold. The increase of temperature from 40 °C to 80 °C of runs SFE1 and SFE2,

respectively, improved the $\eta_{\text{lupenone}}$ by 41% and, if contrasted with the 38% increase verified in the $\eta_{\text{Total}}$ (recall Figure 3), represents a proportional increase. Furthermore, run SFE3 (300 bar, 60 °C, no cosolvent) shows that even though it produced a similar $\eta_{\text{Total}}$ of SFE2, the joint $P - T$ change decreased the lupenone uptake to 2879 mg kg$_{\text{leaves}}^{-1}$. This confirms that the favorable effect of temperature on solubility prevailed over the loss of SC-CO$_2$ density. With the employment of ethanol and ethyl acetate at 300 bar and 80 °C (runs SFE4 and SFE5), higher lupenone yields were obtained, namely 3512 mg kg$_{\text{leaves}}^{-1}$ and 3273 mg kg$_{\text{leaves}}^{-1}$, respectively. This suggests that the joint optimization of $P$, $T$ and cosolvent content can be determinant to potentiate the removal of this compound by SFE.

The low yields for the other triterpenoids explain the lower $\eta_{\text{Total TT}}$ obtained by SFE. These were only slightly inferior to the Soxhlet extractions with *n*-hexane and dichloromethane, but substantially lower in relation to ethyl acetate and ethanol, whose $\eta_{\text{Total TT}}$ values scored, respectively, 74% and 33% higher than of SFE4 (the richest in $\eta_{\text{Total TT}}$ with 4719 mg kg$_{\text{leaves}}^{-1}$).

### 3.4. Triterpenoid Concentration in Extracts

To assess the effect of distinct extraction methods and operating conditions on the selectivity of the identified triterpenoids, their individual concentrations were determined and are presented in Figure 8A,B for Soxhlet and SFE assays, respectively.

Regarding the Soxhlet extracts (Figure 8A), the concentration trend does not follow those of total or individual triterpenoid yields. The highest extract concentration of lupenone amounted 7.7 wt.% and occurred for dichloromethane assay (SX2), followed by *n*-hexane (SX1) with 5.8 wt.%, ethyl acetate (SX3) with 3.8 wt.%, and finally, ethanol (SX4) with 2.2 wt.%. Even though $\eta_{\text{Total TT}}$ and $\eta_{\text{lupenone}}$ of dichloromethane and *n*-hexane Soxhlet extraction were lower than those of ethyl acetate and ethanol, they resulted in higher $C_{\text{lupenone}}$. This is due to the fact that, as discussed previously (see Table 2), the more polar solvents are able to coextract other families of compounds, thus diluting the content of triterpenoids and fading the selectivity towards them, namely to lupenone. The same can be observed for the remaining triterpenoids. Another interesting result is the levelled scores of $C_{\alpha-\text{amyrin}}$ among the four organic solvents studied, ranging from 1.89 to 3.57 wt.%, which were far from expected given the significantly higher $\eta_{\alpha-\text{amyrin}}$ obtained with ethyl acetate and ethanol (see Figure 7A). The observed yield/concentration nuances are ultimately reflected in the $C_{\text{Total TT}}$ response, which was as low as 10.3% and 5.4 wt.% for ethyl acetate and ethanol, respectively, against 13.4 and 14.4 wt.% for dichloromethane and *n*-hexane, respectively. As a result, the use of the two non-polar solvents are better choices than ethanol or ethyl acetate when seeking a higher selectivity to lupenone, despite having lower yields of this compound as counterpart.

The SFE results (see Figure 8B) show higher $C_{\text{lupenone}}$ than any of the Soxhlet extracts, ranging from 10.1 to 12.4 wt.%, for runs SFE5 (300 bar, 80 °C, 5 wt.% of ethyl acetate) and SFE2 (200 bar, 80 °C, no cosolvent), respectively. Once again, one can observe that the discussed trends on $\eta_{\text{Total TT}}$ and $\eta_{\text{lupenone}}$ are not verified for the concentration values. Accordingly, even though run SFE2 showed the highest selectivity towards lupenone, it only attained the third highest $\eta_{\text{lupenone}}$ (see Figure 7B), and the same applies to run SFE1 (200 bar, 40 °C, no cosolvent). In turn, the inclusion of polar modifiers in runs SFE4 (300 bar, 80 °C, 5 wt.% of ethanol) and SFE5 (300 bar, 80 °C, 5 wt.% of ethyl acetate) created the same penalization observed in Soxhlet: lower $C_{\text{lupenone}}$ was attained despite the higher $\eta_{\text{Total TT}}$ and $\eta_{\text{lupenone}}$ scores (see Figures 3 and 7B). As a result, the SFE results show that pure SC-CO$_2$ was the most selective to lupenone, but the method was not able to selectively coextract other triterpenoids. In terms of $C_{\text{Total TT}}$, similarly to what was observed for the triterpenoid yield (see Figure 7B), the values follow the lupenone trend since the remaining triterpenoids were extracted in significantly lower amounts. Accordingly, the maximum was attained by run SFE2 (200 bar, 80 °C, no cosolvent), where $C_{\text{Total TT}}$ is worth 18.0 wt.%. The individual contribution of other triterpenoids in the extract for this score did not surpass 2 wt.%.

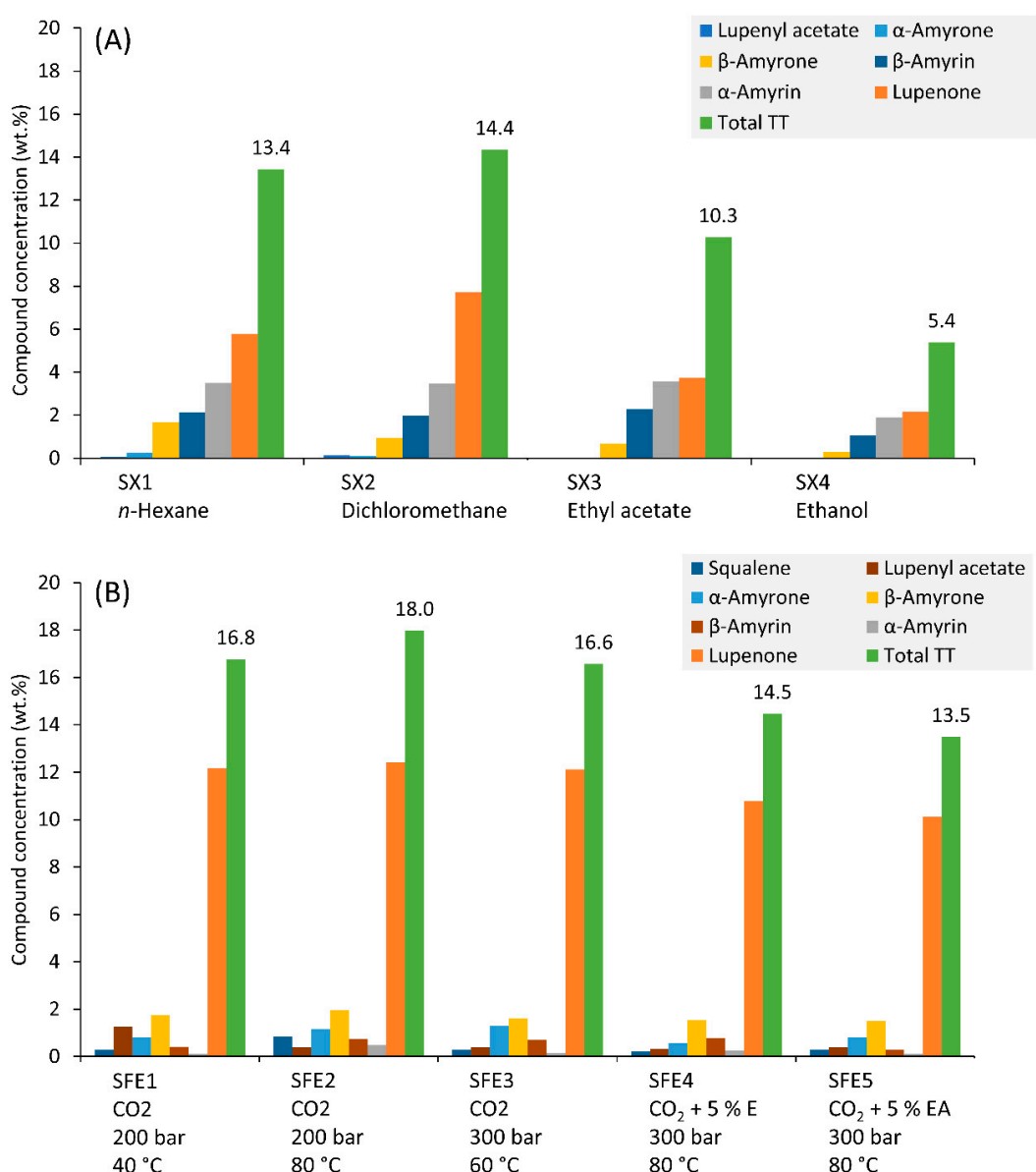

**Figure 8.** Plots of the concentrations of individual triterpenoids identified in (**A**) Soxhlet extracts and respective solvents, and (**B**) SFE at different conditions of temperature, pressure, and addition of ethanol (E) and ethyl acetate (EA) as modifiers.

Overall, the attained SFE results show that the method can selectively extract lupenone from the leaves of *A. dealbata*, even though similar triterpenoids are only coextracted on a negligible basis. Furthermore, lupenone therapeutic potential for inflammation, virus, infection, diabetes, cancer, and treatment of Chagas disease [36] justifies its valorisation in an industrial process for pharmaceutical and food applications.

## 4. Conclusions

Soxhlet extraction of *Acacia dealbata* leaves provided higher total extraction yields than SFE, and the yield increased with the polarity of the organic solvent. In turn, SFE yields were favored by increasing temperature and pressure, and by the addition of polar cosolvents. In both cases the main triterpenoids extracted were squalene, β-amyrone, α-amyrone, β-amyrin, α-amyrin, lupenyl acetate and lupenone, with the latter exhibiting the highest content. Overall, Soxhlet extracts exhibited higher amounts of total triterpenoids in comparison to SFE extracts. Interestingly SFE selectively extracted lupenone, reaching higher contents than those obtained with Soxhlet.

Even though the increase of pressure and temperature combined with the addition of cosolvents favored the SFE total yield, the same effect was not so evident for lupenone yield. In fact, pure $CO_2$ attained a lupenone yield comparable to the maximum value achieved at high pressure with SC-$CO_2$ modified with ethanol. Furthermore, the highest lupenone concentration was also obtained for pure $CO_2$ extracts.

Considering the anti-inflammatory, anti-virus, anti-diabetes, and anti-cancer properties of lupenone, as well as its potential for the treatment of Chagas disease, one may state that SFE contributes to the valorisation of *A. dealbata* leaves in the production of lupenone-enriched extracts for the pharmaceutical, nutraceutical or food industries.

Globally, this work envisions SFE technology as a tool to address the current challenges associated to the management of *A. dealbata* spread, namely by pointing to biorefinery opportunities for its leaves. Even though the present study indicates that high selectivity to lupenone may be achieved by SFE, optimization of the extraction conditions and a preliminary economic evaluation of the process are highly recommended.

**Author Contributions:** Conceptualization, C.M.S.; writing—original draft preparation, V.H.R. and M.M.R.d.M.; methodology, V.H.R. and M.M.R.d.M.; investigation, V.H.R.; formal analysis, M.M.R.d.M., I.P. and C.M.S.; writing—review and editing, I.P. and C.M.S.; supervision, I.P. and C.M.S.; funding acquisition, C.M.S.; resources, C.M.S. All authors have read and agreed to the published version of the manuscript.

**Funding:** This work was developed within the scope of the project CICECO-Aveiro Institute of Materials, UIDB/50011/2020 & UIDP/50011/2020, financed by national funds through the FCT/MEC and, when appropriate, co-financed by FEDER under the PT2020 Partnership Agreement. Authors want to thank Project inpactus—innovative products and technologies from eucalyptus, Project N° 21874 funded by Portugal 2020 through European Regional Development Fund (ERDF) in the frame of COMPETE 2020 n°246/AXIS II/2017. Authors want to thank the funding from Project AgroForWealth (CENTRO-01-0145-FEDER-000001), funded by Centro2020, through FEDER and PT2020.

**Conflicts of Interest:** The authors declare no conflict of interest.

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
