# Peer review of "Extraction of Added-Value Triterpenoids from Acacia dealbata Leaves Using Supercritical Fluid Extraction"

_processes, doi:10.3390/pr9071159_

Round 1

Reviewer 1 Report

The authors write that forest biomass is a byproduct (line 9) of much of it being burned. From the point of view of the RED Directive and RED II of the European Parliament, the wooden biomass is also the wood itself as a construction material which, by being built into buildings, furniture, etc., can constitute a storage place for CO2 captured from the atmosphere. The burning of tree leaves itself is an ineffective process due to the water content. The authors themselves indicate 65,6 % (verse 85). which generates toxic substances as a result of burning. Leaves are most often left in the forest to improve soil properties or composted (biogas can be attempted). This statement makes me miss information in the Introduction what forest biomass is from the legal point of view and what definition the authors used. 

What is missing from the literature review is information on what compounds are observed in plant biomass and what applications they find in industry and what are their uses e.g. in traditional medicine, herbalism. This information is partly found in the conclusions where it should not be. 

Standard deviations are missing from Figures 3,7, and 8. The difference shown could be analyzed with e.g. the Anova test to show the significance of the results obtained. 

The conclusions repeat the assay methodology described in the corresponding section of the article (e.g., lines 357-359). And again the obtained values are given, which were discussed in the chapter results and discussion (3). Information about the determined compounds and their properties is also cited. Information about potential applications of the compounds found in plants (not only in trees and their leaves) should be placed in the Introduction with information where they can be used in industry (also in pharmacological industry). Information type Overall, the main triterpenoid extracted was lupenone, which has several potential therapeutic applications. (line 362). Does not further the knowledge of the possible use and management of the compounds.

Conclusions should include the most important 4-6 insights from the research presented. Without the results that were presented in the chapter above. 

Reviewer 2 Report

This is an interesting paper where authors try to present a new (according to them) aspect, but unfortunately the way of working even correct cannot be considered as a novelty in the field.

So, before publication, I suggest to better underline the new aspects that Authors want to present and to point out how this topic would be of interest.
